# Clinical Importance of Grading Tumor Spread through Air Spaces in Early-Stage Small-Lung Adenocarcinoma

**DOI:** 10.3390/cancers16122218

**Published:** 2024-06-14

**Authors:** Jeong Hyeon Lee, Younggjn Kang, Seojin Kim, Youggi Jung, Jae Ho Chung, Sungho Lee, Eunjue Yi

**Affiliations:** 1Department of Pathology, Korea University Anam Hospital, Seoul 02841, Republic of Korea; pathjhlee@kumc.or.kr (J.H.L.); tamuro@korea.ac.kr (Y.K.); ksjinn@naver.com (S.K.); 2Department of Thoracic and Cardiovascular Surgery, Korea University Anam Hospital, Seoul 02841, Republic of Korea; schifer59@korea.ac.kr (Y.J.); bozof@korea.ac.kr (J.H.C.)

**Keywords:** invasive adenocarcinoma, spread through air spaces, early stage, prognosis

## Abstract

**Simple Summary:**

This study explored how lung cancer spreads through air spaces in small tumors (up to 2 cm) and its impact on patient outcomes. Focusing on small-size and early-stage lung adenocarcinomas, the researchers analyzed medical records of patients treated between 2012 and 2022. We measured the distance of floating cancer cells spread from the main tumor through alveolar spaces and categorized patients based on whether this spread was present and its extent. Interestingly, while overall survival rates were similar across groups, patients with more extensive spread (2 mm or more) experienced a higher chance of cancer recurrence even in this very early cancer. This suggests that understanding the extent of spread through air spaces is crucial in predicting cancer recurrence in small lung tumors. Further research with a larger group of patients is needed to confirm these risk factors and improve treatment strategies.

**Abstract:**

This study aimed to identify the clinical manifestation and implications according to the grading of tumor spread through air spaces in early-stage small (≤2 cm) pathological stage I non-mucinous lung adenocarcinomas. Medical records of patients with pathological stage I tumors sized ≤2 cm were retrospectively reviewed and analyzed. The furthest distance of the spread through air spaces from the tumor margin was measured on a standard-length scale (mm). Enrolled patients were categorized into spread through air spaces (STAS) (−) and STAS (+), and STAS (+) was subdivided according to its furthest distance as follows: STAS (+)-L (<2 mm) and STAS (+)-H (≥2 mm). Risk factors for STAS (+) included papillary predominant subtype (*p* = 0.027), presence of micropapillary patterns (*p* < 0.001), and EGFR (*p* = 0.039). The overall survival of the three groups did not differ significantly (*p* = 0.565). The recurrence-free survival of STAS (+)-H groups was significantly lower than those of STAS (−) and STAS (+)-L (*p* < 0.001 and *p* = 0.039, respectively). A number of alveolar spaces were definite risk factors for STAS (+)-H groups (*p* < 0.001), and male gender could be one (*p* = 0.054). In the patient group with small (≤2 cm) pathological stage I lung adenocarcinomas, the presence of STAS ≥ 2 mm was related to significantly lower recurrence-free survival. For identifying definite risk factors for the presence of farther STAS, more precise analysis from a larger study population should be undertaken.

## 1. Introduction

The existence of isolated tumor cells floating among alveolar spaces separate from the main lesion has been observed for over two decades and has been referred to using different nomenclature in various types of cancers [1]. Two representative studies have established this in lung adenocarcinoma, denoted as tumor spread through air spaces (STAS) [2,3]. It is defined as micropapillary clusters, solid nests, or single cells beyond the edge of the tumor that has spread into air spaces in the surrounding lung parenchyma. This is referred to as the newly introduced pattern of invasion in the 2015 World Health Organization (WHO) classification of lung adenocarcinoma [4]. 

Although STAS is known to be a negative prognostic factor for the recurrence of both non-small cell lung cancer (NSCLC) [5] as well as small cell lung cancer (SCLC) [6] irrespective of the stages, its clinical implication could be particularly pronounced in small size (typically ≤2 m in solid component) pathological stage I lung adenocarcinoma. The potential curative role of limited resection, including segmentectomy and wedge resection [7,8], becomes questionable upon detection of STAS [9]. Numerous studies have reported a significant correlation between higher recurrence rates and the presence of STAS in early-stage lung cancer patients who have undergone limited resection, underscoring the need for careful consideration in surgical decision-making [2,9,10,11,12,13,14]. 

Several studies reported that the presence of STAS still had negative prognostic effects on pathologic stage I lung adenocarcinoma patients who underwent radical resection [15,16,17]. The differences in clinical results from early studies dealing with the correlation between negative prognosis and STAS could be due to the lack of quantitative or qualitative analysis on the morphology of STAS. Several subtypes exist: (1) single cells, (2) micropapillary or ring-like clusters, and (3) solid nests or tumor islands [1,18]. Quantitative differences considering the distance from the edge of the tumor margin or the amount of each STAS could be evaluated. The diversity of STAS seems to be related to recurrence rates and survival among small pathological stage I adenocarcinoma [1,19,20]; therefore, the grading of STAS should be considered in predicting the risk of recurrence. 

Several methods have been designed to estimate the STAS grade. Two methods of measuring the distance from the tumor margin to a commonly used separate tumor island include the standard scale of length (mm) and the number of alveoli in between [2,3]. Moreover, a larger number of tumor clusters has been reported to be associated with worse prognoses. However, to date, no standard grading method has been developed [1,19].

In a previous study, we reported that the presence of STAS was significantly associated with lower recurrence-free survival in patients with early-stage I lung adenocarcinoma treated with standard surgical methods [17]. However, the prognostic influence of the STAS grade on surgically treated early-stage lung adenocarcinomas has not yet been examined. Therefore, this study aimed to investigate how the grading of STAS affects survival outcomes and which variables—either preoperative or histological—could be risk factors for higher grades in surgically treated small pathologic stage I non-mucinous adenocarcinoma of the lung.

## 2. Materials and Methods

### 2.1. Patient Characteristics

We retrospectively reviewed the medical records of patients who underwent surgical resection for lung adenocarcinomas between January 2012 and December 2022 at our institute. Patients with pathological stage I invasive adenocarcinoma ≤ 2 cm in diameter were included in this study. The exclusion criteria were as follows: patients with (1) adenocarcinoma in situ (AIS) or minimally invasive adenocarcinoma (MIA), (2) pathologically confirmed stage II to IV, (3) stage I but invasive tumor size > 2 cm, and (4) mucinous adenocarcinoma (Figure 1). 

The enrolled patients were initially categorized into two groups according to the presence of STAS as follows: STAS (−) and STAS (+). The STAS (+) group was further subdivided according to the furthest distance of STAS from the tumor margin, with STAS (+)-L having a distance of <2 mm and STAS (+)-H having a distance of ≥2 mm. Demographic data, perioperative information, and pathological results were also examined, and survival and risk factor analyses were performed for all three groups. 

This study was approved by the Institutional Review Board of Korea University Anam Hospital (IRB Number 2019AN0146). The necessity for acquiring written informed consent from the patients included in this study was waived because this was a retrospective study, and any individual information was not identifiable in the text. 

### 2.2. Pathological Examination

All pathological slide-archived pathology reports of patients were reviewed by two specialized pulmonary pathologists (Y. Kang and J.H. Lee). These pathologists investigated the presence of STAS and measured the distance and other relevant pathological features, including the micropapillary pattern, lymphovascular invasion or inflammatory reactions, and invasion of the visceral pleura. The clinical and pathological stages of all enrolled patients were revised according to the eighth edition of the American Joint Committee on Cancers (AJCC)/Union for International Cancer Control (UICC) lung cancer staging system [21,22].

STAS was defined based on research by Kadota et al. [4] and is characterized as tumor cells within air spaces (Figure 1). They contain (1) micropapillary or papillary structures without central fibrovascular cores, (2) solid nests or tumor islands consisting of solid collections of tumor cells filling the air spaces, and (3) scattered, discohesive single cells. The distance from the edge of the tumor to the farthest STAS point was determined both by using a ruler and by counting the intervening alveolar spaces. 

The definitions of AIS, MIA, differentiation, measurement of the invasive component, subtypes, and other histologic features were determined by the 2011 adenocarcinoma classification system of the International Association for the Study of Lung Cancer/American Thoracic Society/European Respiratory Society [21].

Tumor sizes were measured in two ways. Before surgery, the largest diameter of the total tumor size and the consolidative part were measured using chest computed tomographic imaging (chest CT). After surgery, the total tumor size and invasive tumor size (excluding the lepidic portion and including only the invasive component) were measured from surgically excised lung specimens by pathologists. The total tumor size was defined as the largest diameter of the tumor, including ground-glass opacity (GGO). The consolidation tumor ratio (CTR) was calculated as the ratio of the consolidative size to the total size, as measured by chest CT. The invasive tumor ratio represents the proportion of the invasive tumor size to the total tumor size. 

### 2.3. Grading of Spread through Air Spaces

The distance from the edge of the tumor to the furthest STAS point was measured using a standard-length scale (mm). A higher grade of STAS denoted as STAS (+)-H, was defined as when the furthest STAS was observed to be 2 mm from the margin. In contrast, if STAS was identified only within the 2 mm boundary, it was included in the lower grade of STAS, namely STAS (+)-L (Figure 2).

### 2.4. Estimation of Clinical Manifestation

All enrolled patients underwent careful physical examination and imaging prior to surgery, and consistent follow-ups in the outpatient department were performed postoperatively. For the initial 2 years post-surgery, patients underwent medical evaluations, including chest CT scans, every 4 months. This frequency shifted to biannual evaluations over the subsequent 3 years.

Recurrence was verified based on the radiological or pathological findings. Locoregional recurrence was defined as the detection of a tumor in the afflicted hemithorax, ipsilateral lung, or ipsilateral mediastinal/hilar lymph nodes, while distant metastasis encompassed any location of recurrence beyond the ipsilateral hemithorax, including the contralateral lung and hilum. Patients who underwent surgery for their lung adenocarcinoma were regularly followed based on the outpatient department and examined according to the follow-up protocols in our institute. The usual follow-up schedule was every 4 months for the first 2 years and then every 6 months for the subsequent 3 years after surgery.

### 2.5. Statistical Analysis

Preoperative clinical variables comprising demographic data as well as findings from chest computed tomography (CT) and pathologic features, including predominant patterns, micropapillary patterns, vascular and lymphovascular invasion, necrosis, and the presence of EGFR, were included in comparison.

For the evaluation of categorical variables, chi-square tests and Fisher’s exact tests (utilized when the expected data value was <5) were employed. Continuous variables were compared using Student’s *t*-test for parametric distributions and Mann–Whitney U tests for nonparametric distributions. ANOVA (Analysis of Variance) test was performed when more than 3 groups were used for analysis. Non-normal continuous variables are summarized with median [IQR or min,max] and Kruskal–Wallis tests.

The logistic regression methods were used to identify variables of statistical significance in relation to the grading of STAS. Consequently, odds ratios (OR) were calculated. The Cox proportional hazard models were performed to identify variables of statistical significance in relation to recurrence. For Multivariate analysis, variables that showed statistical significance (*p*-value < 0.005) were employed in logistic regression methods for identifying significantly related variants for the presence of STAS and Cox’s proportional hazard tests for recurrence risk factors. 

The Kaplan–Meier methods were used for calculating cumulative recurrence-free and overall survival rates. Statistically significant were evaluated using log-rank tests. All survivals were investigated, assuming patients were alive with or without recurrence from the time of surgery. Statistical significances were referred to when the condition of *p*-value was <0.05 with a 5% significance level satisfied. All statistical analyses were conducted using R version 4.3.3 (R Foundation for Statistical Computing, Vienna, Austria).

## 3. Results

### 3.1. Patient Characteristics

In total, 218 patients were enrolled in this study, and STAS was observed in 62 (28.4%). The subgroups of STAS (+), namely STAS (+)-L and STAS (+)-H, comprised 33 (15.1%) and 29 (13.3%) patients, respectively. The mean age at diagnosis was 65.4 ± 9.92 (interquartile range, 59–73) years, while the mean follow-up period was 45.6 ± 24.71 (95% confidential interval, 42.13–48.76) months. 

Clinical stage (*p* < 0.001), including T (*p* = 0.001) factor, consolidative tumor size (*p* < 0.001), and CT ratio (*p* = 0.001), were significantly different between STAS (−) and STAS (+) patients, while only male sex (*p* = 0.005) was significantly different between the STAS (+)-L and -H groups in univariate analysis (Table 1 and Table 2). The demographic and perioperative clinical data of the patients are summarized in Table 1, Table 2 and Appendix A.

### 3.2. Pathological Description

Several histological features, including the predominant subtypes (*p* < 0.001), invasive tumor size (*p* < 0.001) or ratio (*p* < 0.001), presence of micropapillary patterns (*p* < 0.001), pathologic stages (*p* < 0.001) and the presence of EGFR (*p* = 0.006), showed significant differences between STAS (−) and STAS (+) groups (Table 3). No statistically significant variables differed between the STAS (+)-L and STAS (+)-H groups except a number of alveolar spaces (Table 4). The precise pathological features and comparisons among the three groups are summarized in Table 3 and Table 4.

### 3.3. Risk Factors Relating to the Presence and Grading of Spread through Air Spaces

Upon multivariate analysis (Table 5), the papillary predominant subtype (*p* = 0.027), the presence of micropapillary (MP) patterns (*p* < 0.001 and *p* = 0.001 in cases of equal or more than 10% of MP was noted), and the presence of EGFR mutation (*p* = 0.039).

For the presence of STAS more than or equal to 2 mm, the larger number of alveolar spaces or male gender could be possible risk factors (Table 6). The recurrence rates of each group are described in Appendix A.

### 3.4. Survival Analysis

The overall mortality rate in this study was 6.0% (*n* = 13), of which 1.6% (*n* = 4) died of lung cancer. Moreover, the total recurrence rate was 6.4% (*n* = 14; Appendix A). The overall survival did not differ significantly between the STAS (−) and STAS (+) groups (*p* = 0.696) or among the three groups (*p* = 0.565). However, significant differences in recurrence-free survival were observed between the STAS (−) and STAS (+) groups (*p* = 0.001). Those of STAS (+)-H were significantly lower than those of STAS (+)-L (*p* = 0.039) and the other two patient groups combined (*p* < 0.001). The overall survival curves were described in Figure 3 and those of recurrence-free survival in Figure 4. The 3- and 5-year overall as well as recurrence-free survival were described in Appendix A. Uni- and multivariate analysis for recurrence risk factors were in Appendix A.

## 4. Discussion

Currently, STAS is a well-known invasion pattern that has been proven to be a negative prognostic factor for the recurrence of pulmonary malignancy. It was initially adopted to describe the specific histological features of lung adenocarcinoma [4]; however, this could possibly be extended to other types of primary lung cancers, including squamous, small cell, and neuroendocrine tumors [5,12,23].

The association of STAS with poor survival has been reported extensively. In fact, both advanced-stage and appropriately treated early-stage cancers show lower survival rates when combined with STAS [13,19]. Previous studies have revealed that non-anatomical or limited resection could be related to worse prognoses [2,5]. However, several studies have also reported negative clinical outcomes in patients with STAS presenting with early-stage lung cancer treated with standard anatomical resection (lobectomy with mediastinal lymph node dissection) [24].

The two major methods for estimating the STAS include measuring its distance from the edge of the tumor margin using the standard scale of length (mm or μm) and observing the amounts of alveoli in between [2,3,19]. Larger numbers of isolated tumor cells departing from the main lesion would be assumed to have a greater tendency for relapse; however, quantitation has not been successfully established [10,25]. Moreover, although the measurement of the distance from the edge of the tumor margin seems to be more apparent, it has not been standardized [1,19].

The grading system used in our study was based on studies measuring the STAS distance using a standard scale of length [10,19,25]. In previous research, a higher grade of STAS was determined when any existence of tumor cluster ≥ 2500 μm from the margin was observed in all stages of non-small cell lung cancer. Our study cohort was confined to patients with small pathological stage I lung adenocarcinomas, with a measured invasive component of ≤2 cm; therefore, our definition of a higher STAS grade was ≤2 mm, which was also the median value of the STAS distance.

Measuring the number of alveolar spaces between the furthest point of the STAS and the tumor margin offers a more logical point of comparison because direct distance measurement may be affected by suboptimal inflation during the fixation process. Previously, Warth et al. graded STAS based on the intervening alveolar spaces and defined it as ‘limited’ when there were fewer than three spaces and ‘extended’ when there were three or more [3]. In their study, although the overall and recurrence-free survival rates of STAS-positive patients were notably lower than those of STAS-negative patients, no significant differences were discerned between the ‘limited’ and ‘extended’ groups [3]. In our study, the distance of STAS measured by the alveolar spaces showed no significance as a risk factor for recurrence in the multivariate analysis (Table 4), which was similar to our previous research. 

Despite well-established studies proving the clinical implications of STAS [1,19,24,26] and its insignificant relationship with invasive procedures [27,28], criticism remains regarding the origin of STAS. It is believed that STAS can be induced by external mechanical forces, either in vivo or ex vivo. For example, preoperative diagnostic techniques such as percutaneous needle biopsy or any localization method for identifying the precise location of nodules may be involved in the in vivo separation of tumor cells through needle tracks. Moreover, tissue manipulation or palpation during surgery may induce the detachment of loose fragments [29], similar to preoperative diagnostic procedures [30,31]. 

In our study, performing localization procedures using wire before surgery under CT guidance was significantly related to the presence of STAS (*p* = 0.010, Appendix A), while percutaneous needle biopsy was associated with high-grade STAS (*p* = 0.046, Appendix A). Moreover, frozen section analysis showed a higher association with the STAS (−) or low-grade STAS groups (*p* = 0.015, Appendix A). This might suggest that the presence of STAS could potentially be due to pre-surgical stimulation; however, none of these factors were significant risk factors for the presentation of STAS. The higher relevance of the frozen procedure with no or low STAS might be an indirect reflection that STAS could be induced by mechanical forces on unresected malignant lesions rather than from the handling of resected specimens. 

The detachment of malignant loose fragments during the histological tissue handling process also commonly occurs [30,32], in which artifacts are often called STAKS (tumors spread through knife surfaces) [31]. It was previously suspected that the isolated malignant cells could be biologically isolated; however, recent studies using three-dimensional reconstructed microtomographic images have demonstrated that small malignant cell nests can detach from the main lesion, float in the air, reattach to the alveolar wall, and survive [33,34].

In our study, the recurrence-free survival of patients with small, pathological stage I lung adenocarcinomas with high-grade STAS was notably lower, whereas those with negative or low-grade STAS showed no significant differences. In addition, high-grade STAS was more associated with the male sex, larger invasive components, and micropapillary patterns than the other groups. Therefore, although no standardized grading system has been established yet, the measurement of the distance of the furthest STAS point could have significant prognostic predictive value.

The presence of STAS was associated with wild-type EGFR in our study, which is consistent with previous studies [3,35,36]. Although our data on molecular characteristics were limited to EGFR mutations, other molecular features such as KRAS mutation, ALK and ROS1 rearrangements, BRAF mutations, and wild-type HER2 have been reported to be related to the presence of STAS [1,35,37,38]. Several studies have reported no significant relationship between STAS and EGFR mutations [6,10,39].

This study had several limitations. First, this was a retrospective single-center study comprising a small study population. Therefore, the interpretation or implications of the conclusions may be restricted due to inevitable disadvantages such as data selection, confounding, and information bias. Before starting the statistical analysis, we performed power calculations using the chi-square test. In cases of comparison of STAS (−) and STAS (+) group analysis, the result of the test was 1.000. The result of power calculation was also 1.000 for the subdivision of STAS (+) groups. However, the relatively small study population in the subdivision group of STAS (+) might severely restrain the clinical implication of our results. 

Due to the small sample size, the influence of surgical methods (limited or extended, and non-anatomical or anatomical) on survival for each STAS (+) group could not be fully evaluated. All of the enrolled patients underwent VATS (Video-assisted thoracoscopic surgery) procedures, no robotic-assisted procedures were performed, and no open conversions were enforced. The extents of surgical resection along locations were described in Appendix A). No statistical significances were found between STAS subgroups. Therefore, we did not refer to the results. The association between procedures and the presentation of STAS were possibly not fully examined in this study. 

The small study population might be affective for the recurrence risk factor analysis. Visceral pleural invasion, along with the presence of lymphatic, vascular invasion, has been frequently referred to as recurrence risk factors for stage I lung adenocarcinomas [40,41,42]. In our study, no aforementioned factors seemed to have significance both in uni- and multivariate analysis (Appendix A). 

Further prospective studies related to this topic should be designed to achieve more valid results.

Second, the observation periods (mean value of 45.6 ± 24.71 months) for detecting recurrence may not be sufficient to identify differences within each STAS subgroup. Given that all included patients were at pathological stage I, stratified survival analysis with more extensive long-term follow-up is necessary to precisely evaluate the quantitative effects of STAS.

The enrolled patients span over a decade, leading to potential discrepancies in surgical techniques, pathological diagnoses, and medical treatment regimens. Standard VATS procedures have been performed since 2010 at our institutions, and no pulmonary resections were conducted using robotic-assisted thoracoscopic surgery (RATS) during the study period. Thirty-four patients (15.6%) underwent adjuvant chemotherapy, showing no significant statistical differences between STAS subgroups in survival analysis.

The 8th edition of the American Joint Committee on Cancer (AJCC) and Union for International Cancer Control (UICC) lung cancer staging system has been in use since 1 January 2018. Consequently, we revised all the pathological and clinical stages of patients who underwent surgery before 2018 according to the new staging system.

Finally, uncertainties regarding STAS remain, and the discrimination of artifacts depends on the expert skills of pathologists. Although the slides we examined were prepared following standard pathological protocols, including gross inspection, measurement, sectioning, and histological processing, it is possible that STAS could exist in relevant parenchyma that was excluded from sectioning and mapping. Future standard guidelines for detecting STAS, including differential diagnoses from artifacts through immunohistochemistry and molecular analyses, may provide useful tools for identifying this phenomenon. 

Future standard guidelines for detecting STAS and differential diagnoses from artifacts, including immunohistochemistry and molecular analyses, may be useful.

## 5. Conclusions

The presence of STAS equal to or more than 2 mm from the margin of the tumor in small (≤2 cm of invasive component) pathologic stage I lung adenocarcinoma implied significantly lower recurrence-free survival. Further prospective studies comprising a larger study population should be conducted to identify valid prognostic risk factors for those patients. 

## Figures and Tables

**Figure 1 cancers-16-02218-f001:**
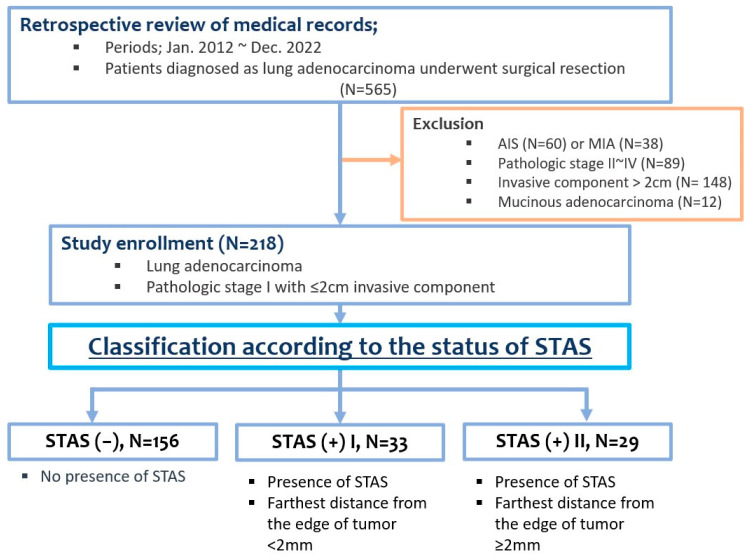
Flowchart of patient enrollment.

**Figure 2 cancers-16-02218-f002:**
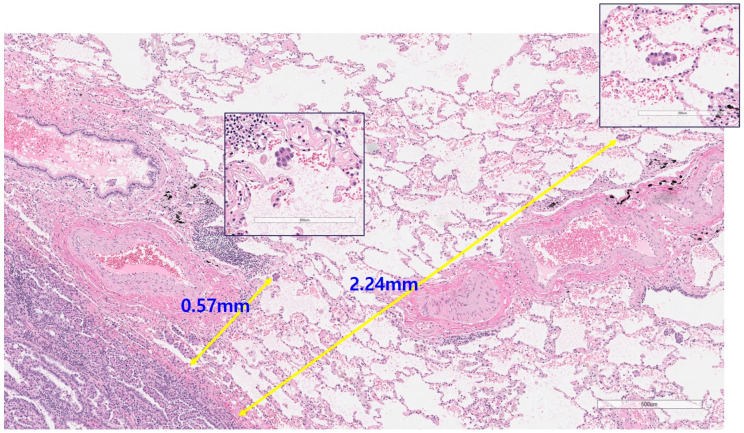
Measurement of the distance from the furthest point of the STAS to the edge of the tumor margin. Low-grade STAS (STAS (+)-L): furthest STAS point of <2 mm. High-grade STAS (STAS (+)-H): furthest STAS point of ≥2 mm. Scale bar: 200 μm (**upper**, **middle**); 500 μm (**bottom**).

**Figure 3 cancers-16-02218-f003:**
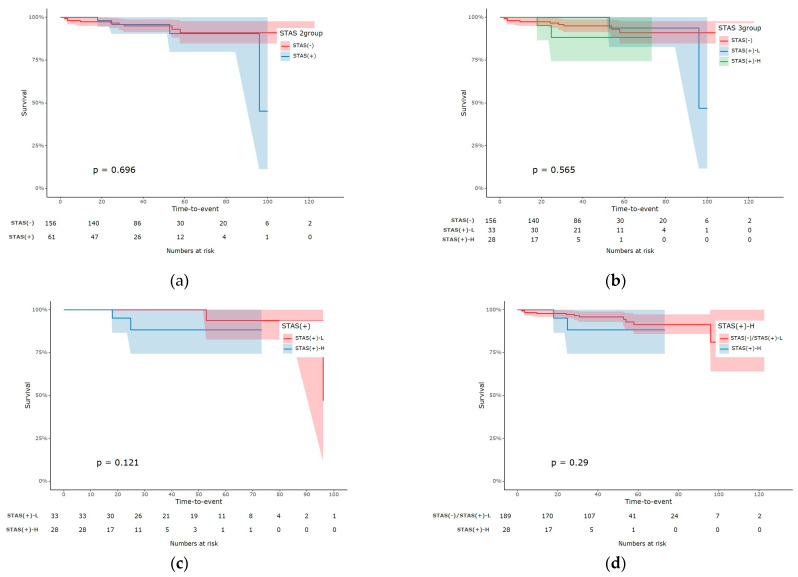
Overall survival (OS) curves. (**a**) OS between STAS (−) and STAS (+), (**b**) OS among three groups, STAS (−), STAS (+)-L, and STAS (+)-H. No significant differences were observed. (**c**) OS between STAS (+)-L and STAS (+)-H, (**d**) OS between STAS (+)-H and other patient groups. No significant differences were identified.

**Figure 4 cancers-16-02218-f004:**
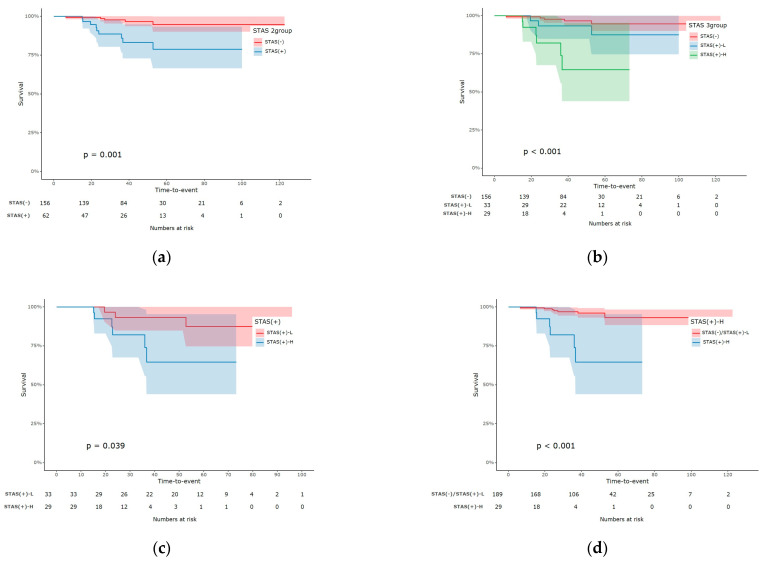
Recurrence-free survival (RFS) curves. (**a**) RFS between STAS (−) and STAS (+), significant differences were observed (*p* = 0.001). (**b**) those among three groups, STAS (−), STAS (+)-L, and STAS (+)-H. (**c**) RFS between STAS (+)-L and STAS (+)-H, (**d**) RFS between STAS (+)-H and other patient groups. STAS (+)-H groups showed significantly lower RFS than other patient groups.

**Table 1 cancers-16-02218-t001:** Demographic and preoperative characteristics of STAS (−) and STAS (+) groups. Clinical staging was determined based on the medical examination, including imaging studies before surgery. Abbreviations: CCI, Charlson Comorbidity Index; CI, confidence interval; CT, computed tomography; CTR, consolidation tumor ratio; cT; cN; GGO, ground glass opacity; PPY, pack per year; SD, standard deviation; STAS, spread through air spaces; SUVmax, maximum standardized uptake value. Data are presented as n(%) and mean ± SD (95% CI).

Variables				STAS (−)	STAS (+)	*p*-Value	Total
				n (%)	n (%)		n (%)
				Mean ± SD (95% CI)	Mean ± SD (95% CI)		Mean ± SD (95% CI)
	**Number**			156 (71.6)	62 (28.4)		218 (100.0)
	**Age**			65.2 ± 1.0 (63.6–66.7)	65.9 ± 9.9 (63.4–68.4)	0.608	65.4 ± 9.9 (64.1–66.7)
						1.000	
			**<65**	68 (43.6)	27 (43.5)		95 (43.6)
			**≥65**	88 (56.4)	35 (56.5)		123 (56.4)
	**Gender**					0.201	
		**Female**		97 (62.2)	32 (51.6)		129 (59.2)
		**Male**		59 (37.8)	30 (48.4)		89 (40.8)
	**CCI**			3.2 ± 1.7 (2.90–3.44)	3.4 ± 2.0 (2.86–3.85)	0.682	3.2 ± 1.8 (2.99–3.46)
						0.878	
		**≤3**		96 (61.5)	37 (59.7)		133 (61.0)
		**>3**		60 (38.5)	25 (40.3)		85 (39.0)
	**Smoking history**					0.344	
			**Never**	109 (69.9)	37 (59.7)		146 (67.0)
			**Current**	21 (13.5)	9 (14.5)		30 (13.8)
			**Ex ≤ 15**	17 (10.9)	12 (19.4)		29 (13.3)
			**Ex > 15**	9 (5.8)	4 (6.5)		13 (6.0)
		**PPY**		9.1 ± 18.1 (6.18–11.91)	16.8 ± 30.6 (8.98–24.52)	0.067	11.2 ± 22.6 (8.22–14.25)
	**Clinical staging**						
		**cT**				0.001	
			**1a**	32 (20.5)	1 (1.6)		33 (15.1)
			**1b**	41 (26.3)	24 (38.7)		65 (29.8)
			**1c**	1 (0.6)	0 (0.0)		1 (0.5)
			**2a**	80 (51.3)	36 (58.1)		116 (53.2)
			**3**	2 (1.3)	1 (1.6)		3 (1.4)
		**cN**				1.000	
			**0**	152 (97.4)	61 (98.4)		213 (97.7)
			**1**	4 (2.6)	1 (1.6)		5 (2.3)
		**Staging**				<0.001	
			**IA1**	31 (19.9)	0 (0.0)		31 (14.2)
			**IA2**	42 (26.9)	24 (38.7)		66 (30.3)
			**IA3**	1 (0.6)	0 (0.0)		1 (0.5)
			**IB**	77 (49.4)	36 (58.1)		113 (51.8)
			**IIA**	3 (1.9)	0 (0.0)		3 (1.4)
			**IIB**	2 (1.3)	2 (3.2)		4 (1.8)
	**CT findings**						
		**Total tumor size (mm, including GGO)**		17.6 ± 5.1 (16.80–18.41)	17.5 ± 3.8 (16.55–18.46)	0.874	17.6 ± 4.7 (16.94–18.21)
		**Consolidative size (mm)**		12.8 ± 5.0 (12.09–13.51)	15.6 ± 2.9 (14.82–16.30)	<0.001	13.6 ± 4.3 (13.01–14.15)
		**CTR**		0.8 ± 0.2 (0.72–0.79)	15.6 ± 2.9 (14.82–16.30)	<0.001	15.6 ± 2.9 (14.82–16.30)
						0.001	
			**0.5≥**	26 (16.7)	1 (1.6)		27 (12.4)
			**0.5<**	130 (83.3)	61 (98.4)		191 (87.6)
		**Pleural tagging**				0.423	
			**N**	74 (47.4)	25 (40.3)		99 (45.4)
			**Y**	82 (52.6)	37 (59.7)		119 (54.6)
		**Location**				0.580	
			**Medial**	19 (12.2)	10 (16.1)		29 (13.3)
			**Lateral**	137 (87.8)	52 (83.9)		189 (86.7)
	**mSUV value**			3.3 ± 3.5 (2.41–4.20)	3.5 ± 2.5 (2.40–4.63)	0.763	3.4 ± 3.2 (2.65–4.07)

**Table 2 cancers-16-02218-t002:** Demographic and preoperative characteristics of STAS (+)-L and STAS (+)-H groups. Abbreviations: CCI, Charlson Comorbidity Index; CI, confidence interval; CT, computed tomography; CTR, consolidation tumor ratio; cT, clinical Tumor stages; cN, clinical Nodal stages ; GGO, ground glass opacity; PPY, pack per year ; SD, standard deviation; STAS, spread through air spaces; SUVmax, maximum standardized uptake value. Data are presented as n (%) and mean ± SD (95% CI).

Variables				STAS (+)-L	STAS (+)-H	*p*-Value	Total
				n (%)	n (%)		n (%)
				Mean ± SD (95% CI)	Mean ± SD (95% CI)		Mean ± SD (95% CI)
	**Number**			33 (53.2)	29 (46.8)		62 (100.0)
	**Age**			65.7 ± 8.5 (62.7–68.7)	66.1 ± 11.4 (61.8–70.5)	0.875	65.9 ± 9.9 (63.4–68.4)
		**Age group**				0.947	
			**<65**	15 (45.5)	12 (41.4)		27 (43.5)
			**≥65**	18 (54.5)	17 (58.6)		35 (56.5)
	**Gender**					0.005	
		**Female**		23 (69.7)	9 (31.0)		32 (51.6)
		**Male**		10 (30.3)	20 (69.0)		30 (48.4)
	**CCI**			3.4 ± 1.8 (2.77–4.08)	3.3 ± 2.1 (2.48–4.08)	0.148	3.4 ± 2.0 (2.86–3.85)
						0.443	
		**≤3**		18 (54.5)	19 (65.5)		37 (59.7)
		**>3**		15 (45.5)	10 (34.5)		25 (40.3)
	**Smoking history**					0.255	
			**Never**	23 (69.7)	14 (48.3)		37 (59.7)
			**Current**	5 (15.2)	4 (13.8)		9 (14.5)
			**Ex ≤ 15**	4 (12.1)	8 (27.6)		12 (19.4)
			**Ex > 15**	1 (3.0)	3 (10.3)		4 (6.5)
		**PPY**		13.8 ± 27.3 (4.09–23.43)	20.2 ± 34.2 (7.15–33.16)	0.423	16.8 ± 30.6 (8.98–24.52)
	**Clinical staging**						
		**cT**				0.505	
			**1a**	1 (3.0)	0 (0.0)		1 (1.6)
			**1b**	14 (42.4)	10 (34.5)		24 (38.7)
			**1c**	0 (0.0)	0 (0.0)		0 (0.0)
			**2a**	17 (51.5)	19 (65.5)		36 (58.1)
			**3**	1 (3.0)	0 (0.0)		1 (1.6)
		**cN**				0.468	
			**0**	33 (100.0)	28 (96.6)		61 (98.4)
			**1**	0 (0.0)	1 (3.4)		1 (1.6)
		**Staging**				0.537	
			**IA1**	0 (0.0)	0 (0.0)		0.0 (0.0)
			**IA2**	15 (45.5)	9 (31.0)		24 (38.7)
			**IA3**	0 (0.0)	0 (0.0)		0.0 (0.0)
			**IB**	17 (51.5)	19 (65.5)		36 (58.1)
			**IIA**	0 (0.0)	0 (0.0)		0 (0.0)
			**IIB**	1 (3.0)	1 (3.4)		2 (3.2)
	**CT findings**						
		**Total tumor size (mm, including GGO)**		17.2 ± 4.1 (15.75–18.68)	17.8 ± 3.4 (16.56–19.11)	0.513	17.5 ± 3.8 (16.55–18.46)
		**Consolidative size (mm)**		14.9 ± 2.8 (13.97–15.91)	16.3 ± 3.0 (15.12–17.40)	0.078	15.6 ± 2.9 (14.82–16.30)
		**CTR**		0.9 ± 0.2 (0.84–0.96)	0.9 ± 0.1 (0.88–0.97)	0.564	0.9 ± 0.2 (0.87–0.95)
						1.000	
			**0.5≥**	1 (3.0)	0 (0.0)		1 (1.6)
			**0.5<**	32 (97.0)	29 (100.0)		61 (98.4)
		**Pleural tagging**				0.536	
			**N**	15 (45.5)	10 (34.5)		25 (40.3)
			**Y**	18 (54.5)	19 (65.5)		37 (59.7)
		**Location**				0.738	
			**Medial**	6 (18.2)	4 (13.8)		10 (16.1)
			**Lateral**	27 (81.8)	25 (86.2)		52 (83.9)
	**mSUV value**			3.9 ± 3.0 (1.75–5.92)	3.2 ± 2.0 (1.86–4.54)	0.561	3.5 ± 2.5 (2.40–4.63)

**Table 3 cancers-16-02218-t003:** Pathologic features and comparison of STAS (−) and STAS (+) groups.

Variables				STAS (−)	STA S (+)	*p*-Value	Total
				n (%)	n (%)		n (%)
				Mean ± SD (95% CI)	Mean ± SD (95% CI)		Mean ± SD (95% CI)
	**Predominant** **subypes**					<0.001	
			**lepidic**	42 (26.9)	1 (1.6)		43 (19.7)
			**acinar**	102 (65.4)	46 (74.2)		148 (67.9)
			**papillary**	4 (2.6)	7 (11.3)		11 (5.0)
			**micropapillary**	0 (0.0)	1 (1.6)		1 (0.5)
			**solid**	8 (5.1)	7 (11.3)		15 (6.9)
	**Tumor size**						
		**Including lepidic component (mm)**		16.6 ± 4.7 (15.89–17.38)	16.9 ± 3.6 (15.95–17.76)	0.710	16.7 ± 4.4 (16.11–17.29)
		**Invasive component (mm)**		12.4 ± 4.2 (11.74–13.07)	15.5 ± 2.7 (14.80–16.12)	<0.001	13.3 ± 4.1 (12.74–13.82)
		**Invasive tumor ratio**		0.77 ± 0.23 (0.735–0.807)	0.93 ± 0.12 (0.903–0.964)	<0.001	0.82 ± 0.22 (0.789–0.846)
	**Visceral pleural invasion**					0.328	
			**0**	73 (46.8)	27 (43.5)		100 (45.9)
			**1**	82 (52.6)	33 (53.2)		115(52.8)
			**2**	1 (0.6)	2 (3.2)		3 (1.4)
	**Micropapillary pattern**					<0.001	
			**No**	145 (92.9)	29 (46.8)		154 (70.6)
			**Yes**	11 (7.1)	33 (53.2)		64 (29.4)
	**Micropapillary pattern (%)**					<0.001	
			**<10**	153 (98.1)	40 (64.5)		193 (88.5)
			**≥10**	3 (1.9)	22 (35.5)		25 (11.5)
	**Lymphovascular invasion**					0.676	
			**No**	150 (96.2)	61 (98.4)		211 (96.8)
			**Yes**	6 (3.8)	1 (1.6)		7 (3.2)
	**Perineural invasion**					1.000	
			**No**	155 (99.4)	62 (100.0)		217 (99.5)
			**Yes**	1 (0.6)	0 (0.0)		1 (0.5)
	**Necrosis**					0.080	
			**No**	151 (96.8)	56 (90.3)		207 (95.0)
			**Yes**	5 (3.2)	6 (9.7)		11 (5.0)
	**Pathologic stage**					<0.001	
			**IA1**	32 (20.5)	0 (0.0)		32 (14.7)
			**IA2**	41 (26.3)	27 (43.5)		68 (31.2)
			**IB**	83 (53.2)	35 (56.5)		118 (54.1)
	**EGFR (+)**					0.006	
			**No**	62 (39.7)	38 (61.3)		100 (45.9)
			**Yes**	94 (60.3)	24 (38.7)		118 (54.1)
**Farthest distance of STAS**							
	**Standard length scale (mm)**				1.9 ± 1.1 (1.60–2.14)		
	**Number of alveolar spaces**				7.2 ± 4.2 (6.12–8.29)		

**Table 4 cancers-16-02218-t004:** Pathologic features and comparison of STAS (+)-L and STAS (+)-H groups.

Variables				STAS (+)-L	STAS (+)-H	*p*-Value	Total
				N (%)	N (%)		N (%)
				Mean ± SD (95% CI)	Mean ± SD (95% CI)		Mean ± SD (95% CI)
	**Predominant subtypes**					0.083	
			**lepidic**	1 (3.0)	0 (0.0)		1 (1.6)
			**acinar**	27 (81.8)	19 (65.5)		46 (74.2)
			**papillary**	1 (3.0)	6 (20.7)		7 (11.3)
			**micropapillary**	1 (3.0)	0 (0.0)		1 (1.6)
			**solid**	3 (9.1)	4 (13.8)		7 (11.3)
	**Tumor size**						
		**Including lepidic component (mm)**		16.6 ± 3.9 (15.26–18.02)	17.1 ± 3.2 (15.87–18.33)	0.608	16.9 ± 3.6 (15.95–17.76)
		**Invasive component (mm)**		15.1 ± 2.7 (14.12–16.00)	16.0 ± 2.7 (14.94–16.70)	0.190	15.5 ± 2.7 (14.80–16.12)
		**Invasive tumor ratio**		0.93 ± 0.14 (0.878–0.975)	0.94 ± 1.00 (0.905–0.978)	0.616	0.82 ± 0.22 (0.789–0.846)
	**Visceral pleural invasion (VPI)**					0.227	
			**0**	16 (48.5)	11 (37.9)		27 (43.5)
			**1**	15 (45.5)	18 (62.1)		33 (53.2)
			**2**	2 (6.1)	0 (0.0)		2 (3.2)
	**Micropapillary pattern**					0.587	
			**No**	17 (51.5)	12 (41.4)		29 (46.8)
			**Yes**	16 (48.5)	17 (58.6)		33 (53.2)
	**Micropapillary pattern (%)**					0.240	
			**<10**	24 (72.7)	16 (55.2)		40 (64.5)
			**≥10**	9 (27.3)	13 (44.8)		22 (35.5)
	**Lymphovascular_invasion**					1.000	
			**No**	32 (97.0)	29 (100.0)		61 (98.4)
			**Yes**	1 (3.0)	0 (0.0)		1 (1.6)
	**Perineural invasion**					1.000	
			**No**	33 (100.0)	29 (100.0)		62 (100.0)
			**Yes**	0 (0.0)	0 (0.0)		0 (0.0)
	**Necrosis**					0.405	
			**No**	31 (93.9)	25 (86.2)		56 (90.3)
			**Yes**	2 (6.1)	4 (13.8)		6 (9.7)
	**Pathologic stage**					0.450	
			**IA1**	0 (0.0)	0 (0.0)		0 (0.0)
			**IA2**	16 (48.5)	11 (37.9)		27 (43.5)
			**IB**	17 (51.5)	18 (62.1)		35 (28.4)
	**EGFR (+)**					0.886	
			**No**	21 (63.6)	17 (58.6)		38 (61.3)
			**Yes**	12 (36.4)	12 (41.4)		24 (38.7)
	**Farthest distance of STAS**						
		**Standard length scale (mm)**		1.0 ± 0.2 (0.93–1.07)	2.9 ± 0.7 (2.60–3.13)	<0.001	1.9 ± 1.1 (1.60–2.14)
		**Number of alveolar spaces**		4.4 ± 1.5 (3.91–4.94)	10.3 ± 4.1 (8.78–11.91)	<0.001	7.2 ± 4.2 (6.12–8.29)

**Table 5 cancers-16-02218-t005:** Multivariate analysis for the presence of STAS (+). Logistic regression method was used with a 95% confidential interval.

Variables			Crude OR (95%CI)	Crude *p*-Value	Adjusted OR (95%CI)	Adjusted *p*-Value
**Smoking History**						
	**PPY**		1.01 (1.00, 1.03)	0.030	1.01 (0.99, 1.03)	0.193
**cT (ref.; 1a)**						
		**1b**	18.73 (2.40, 145.96)	0.005	7.43 (0.7, 79.48)	0.097
		**Ic**	0 (0, Inf)	0.990	0 (0, Inf)	0.994
		**2a**	14.4 (1.89, 109.52)	0.010	9.33 (0.91, 96.14)	0.061
		**3**	16 (0.71, 361.74)	0.081	33.64 (0.38, 3014.12)	0.125
**CT findings**						
	**Consolidative size**		1.17 (1.09, 1.27)	<0.001	1.02 (0.82, 1.26)	0.885
	**CTR**		142.18 (19.8, 1020.95)	<0.001	0.43 (0, 41.41)	0.716
**Pathologic features**						
	**predominant subtype (ref.; lepidic)**					
		**acinar**	18.94 (2.53, 141.86)	0.004	5.71 (0.56, 58.66)	0.143
		**micropapillary**	88,963,559.05 (0, Inf)	0.983	89513922.48 (0, Inf)	0.990
		**papillary**	73.5 (7.13, 757.6)	<0.001	23.77 (1.43, 394.28)	0.027
		**solid**	36.75 (3.96, 340.94)	0.002	8.87 (0.62, 127.05)	0.108
	**Invasive component (mm)**		1.24 (1.14, 1.36)	<0.001	0.98 (0.77, 1.24)	0.856
	**Invasive tumor ratio**		202.39 (20.08, 2039.73)	<0.001	30.44 (0.19, 4777.06)	0.185
	**Micropapillary pattern**		15 (6.81, 33.06)	<0.001	10.94 (3.69, 32.48)	< 0.001
	**Micropapillary pattern (≥10%)**		28.05 (7.99, 98.44)	<0.001	7.4 (1.7, 32.14)	0.008
**EGFR (+)**			0.42 (0.23, 0.76)	0.004	0.38 (0.15, 0.95)	0.039

**Table 6 cancers-16-02218-t006:** Multivariate analysis for the presence of STAS (+)-H over STAS (+)-L. Logistic regression method was used with a 95% confidential interval.

Variables		Crude OR (95%CI)	Crude *p*-Value	Adjusted OR (95%CI)	Adjusted *p*-Value
	**Gender (Male)**	5.11 (1.73, 15.08)	0.003	5.64 (0.97, 32.7)	0.054
	**Number of alveolar spaces**	2.67 (1.57, 4.53)	<0.001	2.78 (1.53, 5.05)	<0.001

## Data Availability

The authors confirm that the data supporting the findings of this study are available in the Appendix A.

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
