# Peer review of "Clinical Importance of Grading Tumor Spread through Air Spaces in Early-Stage Small-Lung Adenocarcinoma"

_cancers, 2024, doi:10.3390/cancers16122218_

Round 1
Reviewer 1 Report
Comments and Suggestions for Authors
The manuscript on the clinical importance of grading tumors through cavity spread (STAS) in early small lung adenocarcinoma provides a valuable contribution to this field.
1. The statistical analysis on page 4 should be numbered 2.5.
2. The second page of the pathological description should be numbered as 3.2.
3. In the second to last line of the abstract, "father" should be changed to "father".
4. Writing errors in STAS (+) - L and STAS (+) - H in Figure 1.
5. In the introduction, it is recommended to briefly outline the pathological and physiological mechanisms of STAS and its role in tumor metastasis, providing readers who are not familiar with the topic with a deeper scientific background. It is recommended to combine previous literature and clarify whether there have been any reports on the relationship between STAS in lung cancer and recurrence rate and survival, with a focus on highlighting the innovative nature of this study and its potential practical clinical significance. Clearly articulate the gaps in current research that your study aims to fill By linking STAS grading with potential impact on patient management, explain how different levels of STAS may affect treatment decisions, such as the scope of surgical resection or the need for adjuvant therapy.
6. In the materials and methods, there is no mention of whether approval has been obtained from the ethics committee.
7. In the exclusion criteria, patients in stages II-IV were excluded, but there were 4 patients in stage II in the tumor staging in Table 1.
8. When discussing limitations, ①the author should discuss whether the sample size is sufficient to detect small but clinically significant differences between groups, especially considering the subdivision into multiple STAS categories. Mention whether power calculations were performed and provide results to support the sample size used ;②Discuss whether the follow-up period is sufficient to capture all relevant clinical outcomes, especially recurrence, which varies greatly in time. If the follow-up time is relatively short, some late stage relapses may not be detected, which may lead to biased results ;③The article did not elaborate in detail on how changes in surgical techniques, adjuvant therapy, or other treatment decisions may affect outcomes ;④Given that the data spans over a decade, changes in diagnostic techniques, surgical techniques, and pathological evaluation methods may affect the results of the study over time.
Author Response
RESPOND TO THE REVIEWERS
The authors would like to thank Reviewers for careful review of our manuscript and providing us with their comments and suggestion to improve the quality of the manuscript. The following responses have been prepared to address all of the editor’s comments in a point –by-point fashion.
Reviewer 1:
The manuscript on the clinical importance of grading tumors through cavity spread (STAS) in early small lung adenocarcinoma provides a valuable contribution to this field.
Comment 1
: The statistical analysis on page 4 should be numbered 2.5.
Answer 1
: I and my coauthors are deeply grateful for your comments, and are terribly sorry for the mistakes. We noticed that the number of the subtitle ‘statistical analysis’ was missed, therefore added the number 2.5 as the reviewer had pointed out.
Correction 1
: We identified there was a missing number in the subtitle of ‘Statistical analysis’ in the section of Materials and Methods. We corrected the errors in page 4 according to the reviewer’s comments. Changes were marked with yellowish color in the text.
The missing number of Statistical analysis in page 4 was added as follows;
2.5. Statistical analysis
Comment 2
: The second page of the pathological description should be numbered as 3.2.
Answer 2
: We sincerely apologize for the awful mistakes again. And we would like to express our deepest gratitude for the thoughtful reviewing of our manuscript. We identified as well as corrected the errors as the reviewer had pointed out.
Correction 2
: We checked and corrected the numbering errors of the subtitle ‘pathologic description’ in the section of Results according to the reviewer’s mention. Corrections were highlighted in yellowish color in the text.
In page 7, the number of 3.1. Pathological description was corrected as follows;
3.2. Pathological description
Comment 3
: In the second to last line of the abstract, "father" should be changed to "farther".
Answer 3
: I and my coauthors are grateful for your careful comments. We are also terribly sorry for the repetitive mistakes. We checked the spelling errors that the reviewer had pointed out, and corrected them as the reviewer had suggested.
Correction 3
: in the second to the last line of the abstract, the word “father” was changed into “farther” according to the reviewer’s comments. Corrections were highlighted in yellowish color in the text.
For identifying definite risk factors for the presence farther STAS, more precise analysis from larger study population should be proceeded.
Comment 4
: Writing errors in STAS (+) - L and STAS (+) - H in Figure 1.
Answer 4
: Thank you very much for the detection of describing errors in the Figure 1. And we sincerely for the severe mistakes. We identified the wrong description of STAS (+) groups as well as corrected them STAS (+)-L and STAS (+)-H in Figure 1 according to the reviewer’s advice.
Correction 4
: We corrected the writing errors in the Figure 1 as the reviewer had pointed out. The ligand of changed figure was marked with yellowish color in the text.
Figure 1. Flowchart of patient enrollment
Comment 5
: In the introduction, it is recommended to briefly outline the pathological and physiological mechanisms of STAS and its role in tumor metastasis, providing readers who are not familiar with the topic with a deeper scientific background. It is recommended to combine previous literature and clarify whether there have been any reports on the relationship between STAS in lung cancer and recurrence rate and survival, with a focus on highlighting the innovative nature of this study and its potential practical clinical significance. Clearly articulate the gaps in current research that your study aims to fill by linking STAS grading with potential impact on patient management, explain how different levels of STAS may affect treatment decisions, such as the scope of surgical resection or the need for adjuvant therapy.
Answer 5
: I and my coauthors are really grateful for the reviewer’s careful comment. We do understand we should describe the characteristics and clinical implication of STAS, as well as the novel aspect of our study in the section of introduction. We appended the description relating with clinical implication and our research object in the Introduction according to the reviewer’s advice. Changes were marked with yellowish color in the text.
Correction 5
: We added several paragraphs describing the clinical implication and our past research, as well as the purpose of our current study in the section of Introduction as the reviewer’s comment. Corrections were marked with yellowish color in the text.
Although STAS is known to be a negative prognostic factor for the recurrence of both non-small cell lung cancer (NSCLC) [5] as well as small cell lung cancer (SCLC) [6] irrespective of the stages, its clinical implication could be particular pronounced in small size (typically ≤2m in solid component) pathological stage I lung adenocarcinoma. The potential curative role of limited resection, including segmentectomy and wedge resection [7, 8], becomes questionable upon detection of STAS [9]. Numerous studies have reported a significant correlation between higher recurrence rates and the presence of STAS in early-stage lung cancer patients who have under-gone limited resection, underscoring the need for careful consideration in surgical decision-making [2, 9-14].
Several researches reported that the presence of STAS still had negative prognostic effects on pathologic stage I lung adenocarcinoma patients who underwent radical resection [15-17]. The differences in clinical results from early researches dealing with the correlation between negative prognosis and STAS could be due to the lack of quantitative or qualitative analysis on the morphology of STAS. Several subtypes exist; (1) single cells, (2) micropapillary or ring-like clusters and (3) solid nests or tumor islands [1, 18]. Quantitative differences considering distance from the edge of tumor margin or amount of each STAS could be evaluated. Those diversity of STAS seems to be related with recurrence rates and survivals among in small pathological stage I adenocarcinoma. [1, 19, 20]; therefore, the grading of STAS should be considered in predicting the risk of recurrence.
Several methods have been designed to estimate the STAS grade. Two methods of measuring the distance from the tumor margin to a commonly used separate tumor island include the standard scale of length (mm) and the number of alveoli in be-tween [2, 3]. Moreover, a larger number of tumor clusters has been reported to be as-sociated with worse prognoses. However, to date, no standard grading method has been developed [1, 19].
In a previous study, we reported that the presence of STAS was significantly as-sociated with lower recurrence-free survival in patients with early stage I lung adenocarcinoma treated with standard surgical methods [17]. H, the prognostic influence of the STAS grade on surgically treated early stage lung adenocarcinomas has not yet been examined. Therefore, this study aimed to investigate how the grading of STAS affects survival outcomes and which variables—either preoperative or histological—could be risk factors for higher grades in surgically treated small pathologic stage I non mucinous adenocarcinoma of the lung.
Comment 6
: In the materials and methods, there is no mention of whether approval has been obtained from the ethics committee.
Answer 6
: Thank you very much for the reviewer’s comment. The approval from the ethics committee of institute is one of the most importance process which should be performed before starting studies containing patient data. The protocol of this study and a waiver of informed consent was approved by the Institutional Review Board of Korea University Anam Hospital (IRB Number 2019AN0146).
All methods including retrospective data collection and analysis were performed in accordance with relevant Korean guidelines and regulations. We added the sentences relating with ethical approval at the end of 2.1. Patient characteristics in the section of Materials and Methods according to the reviewer’s advice.
Correction 6
: We appended sentences describing ethical approval of or study at the end of 2.1. Patient characteristics in the section of Materials and Methods as the reviewer’s comment. Changes were marked with yellowish color in the text.
This study has been approved by the Institutional Review Board of Korea University Anam Hospital (IRB Number 2019AN0146). The necessity for acquiring writ-ten informed consent from the patients included in this study was waived because this was retrospective study and any individual information was not identifiable in the text.
Comment 7
: In the exclusion criteria, patients in stages II-IV were excluded, but there were 4 patients in stage II in the tumor staging in Table 1.
Answer 7
: We sincerely appreciate for the reviewer’s comment. The study population of our study was pathologically diagnosed stage I lung adenocarcinoma with tumor size ≤2cm. Therefore, patients who were diagnosed as pathological stage II-IV (patient who were proved to have hilar or mediastinal lymph node metastasis or distant metastasis, or adjacent organ metastasis). Patients should undergo serial medical examination including imaging studies for deciding clinical staging. The clinical staging is useful for deciding which treatment strategy is appropriate for each patient, however, it is not confirmative. Pathologic stages could be determined after precise pathologic examination from surgically extracted specimen. Patient who had clinical stage II because of the possible hilar lymph node metastasis in imaging study could be determined as stage I when the suspected hilar lymph node was confirmed to be negative after surgery. In our study, some of our including patients had clinical stage II, and they were confirmed to be pathological stage I after surgery by precise pathologic examination of their surgical specimen.
The Contents in Table 1 described clinical data of study patients before surgery, therefore we used their clinical stages. We should have more carefully express differences between clinical and pathologic stages in the Table 1. We do apologize for the ambiguity and would like to complement the ligand of Table 1 as the reviewer’ comment.
Correction 7
: We added description relating with the clinical staging in the ligand of Table 1. according to the reviewer had pointed out. Corrections were marked with yellowish color in the text.
Table 1. Demographic and preoperative characteristics of STAS (-) and STAS (+) groups. Clinical staging was determined based on the medical examination including imaging studies before surgery.
Comment 8
: When discussing limitations, ①the author should discuss whether the sample size is sufficient to detect small but clinically significant differences between groups, especially considering the subdivision into multiple STAS categories. Mention whether power calculations were performed and provide results to support the sample size used; ②Discuss whether the follow-up period is sufficient to capture all relevant clinical outcomes, especially recurrence, which varies greatly in time. If the follow-up time is relatively short, some late stage relapses may not be detected, which may lead to biased results; ③The article did not elaborate in detail on how changes in surgical techniques, adjuvant therapy, or other treatment decisions may affect outcomes; ④ Given that the data spans over a decade, changes in diagnostic techniques, surgical techniques, and pathological evaluation methods may affect the results of the study over time.
Answer 8
: I and all of my coauthors would like to express our deepest gratitude for the reviewer’s valuable comment. We totally agreed that we should have discussed the points that the reviewer had suggested before we submitted our manuscript. We revised our manuscript and appended the following contents as the reviewer’s advice. Changes were marked with yellowish color in the text.
Correction 8
: About discussion in limitation ①, we revised the statistical analysis and performed power calculation. Those calculation for each group was performed based on chi-square tests. And we added description relating with the power calculation at the end of the Discussion. Corrections were highlighted with yellowish color in the text.
Therefore, the interpretation or implications of the conclusions may be restricted due to inevitable disadvantages such as data selection, confounding, and information bias. Before starting statistical analysis, we had performed power calculation using chi-square test. In cases of comparison of STAS (-) and STAS (+) group analysis, the result of the test was 1.000. The result of power calculation was also 1.000 for subdivision of STAS (+) groups. However, relatively small study population in subdivision group of STAS (+) might severely restrain the clinical implication of our results.
Due to the small sample size, the influence of surgical methods (limited or ex-tended, and non-anatomical or anatomical) on survival for each STAS (+) groups could not be fully evaluated. Therefore, the association between procedures and presentation were possibly not fully examined in this study. Further prospective studies related to this topic should be designed to achieve more valid results.
: About discussion in limitation ② we appended the sentences describing limitations due to the relatively short-term follow-up periods as the reviewer’s advice. Also, we added the follow-up protocols in 2.4 Estimation of clinical manifestation in the section of Materials and Methods. Changes were marked with yellowish color in the text.
In the section of Materials and Methods,
2.4 Estimation of clinical manifestation
Patients who underwent surgery for their lung adenocarcinoma were regularly followed based on the outpatient department, and examined according to the follow-up protocols in our institute. The usual follow-up schedule was every 4 months for the first 2 years and then every 6 months for the subsequent 3 years after surgery.
In the section of Discussion,
Second, the observation periods (mean value of 45.6 ± 24.71 months) for detecting recurrence may not be sufficient to identify differences within each STAS subgroup. Given that all included patients were at pathological stage I, a stratified survival analysis with more extensive long-term follow-up is necessary to precisely evaluate the quantitative effects of STAS.
: About discussion in limitation ③, we appended descriptions relating with surgical methods as limitation in the section of discussion.
Due to the small sample size, the influence of surgical methods (limited or extended, and non-anatomical or anatomical) on survival for each STAS (+) groups could not be fully evaluated. All of the enrolled patient underwent VATS (Video-assisted thoracoscopic surgery) procedures, no robotic assisted procedures were performed, and no open conversion were enforced. The extents of surgical resection along locations were described in Supplemental materials (Table S3). No statistical significances were found between STAS subgroups, therefore we did not refer the results. The association between procedures and presentation of STAS were possibly not fully examined in this study. Further prospective studies related to this topic should be designed to achieve more valid results.
: About discussion in limitation ④, we appended descriptions relating with surgical methods as limitation in the section of discussion. Also, we added the sentences describing revision of pathologic and clinical stages were added. Changes were marked with yellowish color in the text.
In 2.2. Pathological examination of the section of Materials and Methods,
Clinical and pathological stages of all enrolled patients were revised according to the eighth edition of the American Joint Committee on Cancers (AJCC)/Union for Inter-national Cancer Control (UICC) lung cancer staging system [21, 22].
In the section of Discussion, we added sentences regarding limitations due the extensive study periods.
The enrolled patients span over a decade, leading to potential discrepancies in surgical techniques, pathological diagnoses, and medical treatment regimens. Stand-ard VATS procedures have been performed since 2010 at our institutions, and no pulmonary resections were conducted using robotic-assisted thoracoscopic surgery (RATS) during the study period. Thirty-four patients (15.6%) underwent adjuvant chemotherapy, showing no significant statistical differences between STAS sub-groups in survival analysis.
The 8th edition of the American Joint Committee on Cancer (AJCC) and Union for International Cancer Control (UICC) lung cancer staging system has been in use since January 1, 2018. Consequently, we revised all the pathological and clinical stages of patients who underwent surgery before 2018 according to the new staging system. Although the 9th edition was launched in 2024, the T descriptions remain un-changed, and all of our study population were N0. Therefore, we did not revise the staging according to the new system.
The authors sincerely appreciate the editor’s and reviewers’ valuable comments. We carefully revised our manuscript according to the reviewers’ precious advice. We believe we did our best to improve the quality of our manuscript, and wish our revision have better achievement. If there were anything to be corrected or appended, please let us know and we promise we will make every effort to revise again.

Reviewer 2 Report
Comments and Suggestions for Authors
The manuscript was well writen with acceptable language and the layout was fine. The research is of novelty. However, there are several defects in the study design leading to the consideration of publication doubtful. The following are my comments.
(1) The histologies of the studied tumor included micropapillary, papillary, solid, lepidic, acinar etc. As we know the sensitivity of inducing STAS are different amongst various pathological types. The most will be micropapillary type. I wonder if it is suitable, in your study, to count them as one.
(2) How much time you had to spend on measuring the STAS farthest distance and the number of alveolar space in each case ?
(3) A very important question is the accurate measurement of the STAS farthest distance. Did the pathologists use the whole slide imaging ? If not, how did you know you choose the "real" one ?
(4) As you stated in your text, mechanical force is a very important factor affecting the STAS. Since for the modern thoracic surgery, the incisions are made smaller and smaller. The specimens are always taken out through a very narrow wound so that they are always squeezed. Also there will be much discrepancies among surgeons on their dexterity. That will be a big problem.
Author Response
RESPOND TO THE REVIEWERS
The authors would like to thank Reviewers for careful review of our manuscript and providing us with their comments and suggestion to improve the quality of the manuscript. The following responses have been prepared to address all of the editor’s comments in a point –by-point fashion.
Reviewer 2
The manuscript was well written with acceptable language and the layout was fine. The research is of novelty. However, there are several defects in the study design leading to the consideration of publication doubtful. The following are my comments.
Comment 1
: The histologies of the studied tumor included micropapillary, papillary, solid, lepidic, acinar etc. As we know the sensitivity of inducing STAS are different amongst various pathological types. The most will be micropapillary type. I wonder if it is suitable, in your study, to count them as one.
Answer 1
: We sincerely appreciated for the reviewer’s careful advice. There are several representative predominant subtypes; lepidic, acinar, papillary, micropapillary and solid patterns. These predominant subtypes were based on the International Association for the Study of Lung Cancer/American Thoracic Society/European Respiratory Society International Multidisciplinary Classification of Lung Adenocarcinoma which were published in 2011.
Micropapillary patterns are characteristic pathological findings consisting of unique growth patterns of small papillary clusters of glandular cells growing within the airspace (Cakir E, Yilmaz A, Demirag F, et al. Prognostic significance of micropapillary pattern in lung adenocarcinoma and expression of apoptosis-related markers: caspase-3, bcl-2, and p53. APMIS. 2011;119:574-80). This pattern is noted in micropapillary predominant pattern subtype adenocarcinomas, however, could be identified in all subtypes of invasive adenocarcinomas. The presence of micropapillary pattern in a tumor does not mean that it belongs to the histologic subtype of micropapillary predominant patterns. The presence of micropapillary patterns were known to be closely related with the presence of STAS (Kadota K, et al. Tumor Spread Through Air Spaces is an Important Pattern of Invasion and Impacts the Frequency and Location of Recurrences Following Limited Resection for Small Stage I Lung Adenocarcinomas. J Thorac Oncol. 2015;10:806-14.) , and also as in independent prognostic factors for recurrence of lung adenocarcinomas (Nitadori, Jun-ichi, et al. "Impact of micropapillary histologic subtype in selecting limited resection vs lobectomy for lung adenocarcinoma of 2cm or smaller." Journal of the National Cancer Institute 105.16 (2013): 1212-1220.) The presence of micropapillary patterns were usually described as percentages in pathologic reports.
Comment 2
: How much time you had to spend on measuring the STAS farthest distance and the number of alveolar spaces in each case?
Answer 2
: Thank you very much for the reviewer's comments. The time required for detecting STAS depends on the subtypes and sizes of the tumors, including the presence of a lepidic component. The number of pathological slides prepared for diagnosis in each case varies from several to dozens, with more required if special immunohistologic studies are necessary. For relatively straightforward cases, such as adenocarcinoma with acinar predominant subtypes lacking a lepidic component and without visceral pleural invasion, a few minutes may suffice. However, for more complex cases, such as those involving acinar predominant subtypes with micropapillary or solid components and visceral pleural invasion, additional slides are required for a more definitive diagnosis, which can sometimes take hours. For the retrospective review and data collection of this study, we retrieved all related pathological slides, averaging 16 slides per case. Two pathologists, J.H. Lee and Y. Kang, examined and evaluated the presence of STAS.
Comment 3
: A very important question is the accurate measurement of the STAS farthest distance. Did the pathologists use the whole slide imaging? If not, how did you know you choose the "real" one?
Answer 3
: We are really grateful for the reviewer’s comment. For evaluating the clinical implication of STAS, it is very important that the detected STAS should be real one. There were two published studies suggesting that STAS might partly be due to artifacts caused by a prosecting knife during specimen handling (Thunnissen E, Blaauwgeers HJ, de Cuba EM, et al. Ex vivo artifacts and histopathologic pitfalls in the lung. Arch Pathol Lab Med (2016) 140:212–20., Blaauwgeers H, Flieder D, Warth A, Yick CY, Flieder DB. A prospective study of loose tissue fragments in non-small cell lung cancer resection specimens: an alternative view to “Spread Through Air Spaces”. Am J Surg Pathol (2017) 41:1226–30). Our pathologists pay careful attention during specimen handling not to induce artifacts caused by a prosecting knife. To reduce influence of knife on the presence of STAS, our pathologists used the following protocols: First, the lung cancer specimen was cut at its largest diameter using a clean, long prosecting knife, second the specimens were cut along the vertical direction of the first cut by using a second clean knife, and last, all specimens were cut in a single continuous direction to avoid excessive tumor tissue contamination caused by drawing the knife back and forth. Single cell nests or single cell clusters near the knife-cutting surfaces between two blocks are likely to be artifacts so we excluded and did not count as the presence of STAS.
Comment 4
: As you stated in your text, mechanical force is a very important factor affecting the STAS. Since for the modern thoracic surgery, the incisions are made smaller and smaller. The specimens are always taken out through a very narrow wound so that they are always squeezed. Also, there will be much discrepancies among surgeons on their dexterity. That will be a big problem.
Answer 4
: Thank you very much for your invaluable comments. In minimally invasive surgery, manipulation through a narrow surgical wound can lead to the detachment of loose cancer tissue fragments. Preoperative invasive diagnostic procedures, such as percutaneous needle biopsy, bronchoscopy, or CT-guided localization, may contribute to this phenomenon.
In our study, we demonstrated that the performance of localization procedures using wire before surgery under CT guidance was significantly associated with the presence of STAS (p=0.010, Table S1). Additionally, percutaneous needle biopsy was correlated with high-grade STAS (p=0.046, Table S1). Furthermore, frozen section analysis showed a higher association with the STAS (-) or low-grade STAS groups (p=0.015, Table S3), suggesting that the presence of STAS could potentially be attributed to pre-surgical stimulation. However, no significant relationships were observed between the presence of STAS and the examined factors. Although we compared perioperative factors, including surgical methods and tumor location, no statistical significances were found. We thought that this would be due relatively small study population and described as limitation in the section of Discussion.
Correction 4
: We described the limitation relating with surgical methods in the section of Discussion according to the reviewer’s comment. Changes were highlighted with yellowish in the text.
Due to the small sample size, the influence of surgical methods (limited or ex-tended, and non-anatomical or anatomical) on survival for each STAS (+) groups could not be fully evaluated. All of the enrolled patient underwent VATS (Video-assisted thoracoscopic surgery) procedures, no robotic assisted procedures were performed, and no open conversion were enforced. The extents of surgical resection along locations were described in Supplement Table S3. No statistical significances were found between STAS subgroups, therefore we did not refer the results. The as-sociation between procedures and presentation were possibly not fully examined in this study. Further prospective studies related to this topic should be designed to achieve more valid results.
The authors sincerely appreciate the editor’s and reviewers’ valuable comments. We carefully revised our manuscript according to the reviewers’ precious advice. We believe we did our best to improve the quality of our manuscript, and wish our revision have better achievement. If there were anything to be corrected or appended, please let us know and we promise we will make every effort to revise again.

Reviewer 3 Report
Comments and Suggestions for Authors
This is a retrospective study investigating the importance of tumor spread in air spaces in patients suffering from lung cancer.
This is a study with promising results, but there are a few considerable issues.
1. The authors must explain the rationale for including T1a and T1b, but excluding T1c (2cm < T < 3cm). It seems logical to include all T1 patients (nodular lesions) and only exclude T2 (mass). Please explain.
2. In the patient characteristics section you refer to "pathologically confirmed stages II to IV". I suppose you mean staging II to IV, otherwise in stage IV for example you need pathologic specimens from metastatic regions.
3. TNM staging version 7 was in effect for Lung Cancer until 2018. Then version 8 was introduced. Did you use the last version for restaging older patients for the needs of the study? Please explain.
4. In the methods section, the authors describe post-surgery remaining tumor. This does not make sense, because IA lung cancer can be treated with surgery, without any tissue remaining. Please explain.
5. The authors have only attempted correlations with EGFR mutations. Please explain the rationale for not performing ALK or PD-L1 analysis as well.
6. Please make sure to have a cross-check with clinical colleagues for consistency of terms.
Please make sure to have a cross-check with clinical colleagues for consistency of terms.
Author Response
RESPOND TO THE REVIEWERS
The authors would like to thank Reviewers for careful review of our manuscript and providing us with their comments and suggestion to improve the quality of the manuscript. The following responses have been prepared to address all of the editor’s comments in a point –by-point fashion.
Reviewer 3
This is a retrospective study investigating the importance of tumor spread in air spaces in patients suffering from lung cancer. This is a study with promising results, but there are a few considerable issues.
Comment 1
: The authors must explain the rationale for including T1a and T1b, but excluding T1c (2cm < T < 3cm). It seems logical to include all T1 patients (nodular lesions) and only exclude T2 (mass). Please explain.
Answer 1
: I and my coauthors sincerely appreciate for the reviewer’s valuable comment. We excluded cancers with invasive tumor size larger than 2cm in diameter among pathologic stage I lung adenocarcinomas. Although the presence of STAS has been proved to be a negative prognostic factor for non-small cell and small cell lung cancers regardless of stages, it could be more important in small size cancers, typically ≤2m in largest diameter. Tumors larger than 2cm are not likely to be candidates for limited resection. Studies evaluating the relationship between STAS and early-stage lung cancers usually enroll patients with ≤2m in diameter, therefore we would like to follow the trend. We added the explanation regarding with including small size and pathologic stage I lung adenocarcinoma in the section of Introduction as the reviewer’s advice.
Correction 1
: We appended sentences describing including patients with small size (typically ≤2m in largest diameter) of pathologic stage I lung adenocarcinoma in the section of Introduction according to the reviewer’s comments. Changes were highlighted with yellowish color in the text.
Although STAS is known to be a negative prognostic factor for the recurrence of both non-small cell lung cancer (NSCLC) [5] as well as small cell lung cancer (SCLC) [6] irrespective of the stages, its clinical implication could be particular pronounced in small size (typically ≤2m in solid component) pathological stage I lung adenocarcinoma. The potential curative role of limited resection, including segmentectomy and wedge resection [7, 8], becomes questionable upon detection of STAS [9]. Numerous studies have reported a significant correlation between higher recurrence rates and the presence of STAS in early-stage lung cancer patients who have under-gone limited resection, underscoring the need for careful consideration in surgical decision-making [2, 9-14].
Several researches reported that the presence of STAS still had negative prognostic effects on pathologic stage I lung adenocarcinoma patients who underwent radical resection [15-17]. The differences in clinical results from early researches dealing with the correlation between negative prognosis and STAS could be due to the lack of quantitative or qualitative analysis on the morphology of STAS. Several subtypes exist; (1) single cells, (2) micropapillary or ring-like clusters and (3) solid nests or tumor islands [1, 18]. Quantitative differences considering distance from the edge of tumor margin or amount of each STAS could be evaluated. Those diversity of STAS seems to be related with recurrence rates and survivals among in small pathological stage I adenocarcinoma. [1, 19, 20]; therefore, the grading of STAS should be considered in predicting the risk of recurrence.
Comment 2
: In the patient characteristics section you refer to "pathologically confirmed stages II to IV". I suppose you mean staging II to IV, otherwise in stage IV for example you need pathologic specimens from metastatic regions.
Answer 2
: Thank you very much for the reviewer’s comment. In describing exclusion criteria, we referred “pathologically confirmed stage II to IV, which meant, we excluded patients who proved to node positive or distant metastasis after surgical resection. The retrospective data we had reviewed for this study were all from the patients who underwent surgery for their cancers, we confirmed metastasis from surgically resected specimens.
Comment 3
: TNM staging version 7 was in effect for Lung Cancer until 2018. Then version 8 was introduced. Did you use the last version for restaging older patients for the needs of the study? Please explain.
Answer 3
: We are truly grateful the reviewer’s comment. The 8th AJCC/UICC staging system for lung adenocarcinomas had been launched since the 1st January 2018. We reviewed all the clinical and pathologic data of patients who underwent surgical resection before 2018 and revised the stages according to the 8th edition. The new 9th AJCC/UICC system has been started from 2024, however, the T descriptions remains unchanged, and all our study population were N0, therefore we did not renew the stages according to the new staging system. We appended the description relating with staging in 2.2 Pathologic examination in the section of Materials and Methods as well as Discussion according to the reviewer’ advice. Corrections were marked with yellowish color in the text.
Correction 3
: in 2.2 Pathologic examination in the section of Materials and Methods, we added sentences describing revising clinical and pathologic stages of the enrolled patients. Changes were marked with yellowish color in the text.
Clinical and pathological stages of all enrolled patients were revised according to the eighth edition of the American Joint Committee on Cancers (AJCC)/Union for Inter-national Cancer Control (UICC) lung cancer staging system [21, 22].
[21] Travis, W.D., et al., The IASLC lung cancer staging project: proposals for coding T categories for subsolid nodules and assessment of tumor size in part-solid tumors in the forthcoming eighth edition of the TNM classification of lung cancer. Journal of Thoracic Oncology, 2016. 11(8): p. 1204-1223.
[22; Edition, S., S. Edge, and D. Byrd, AJCC cancer staging manual. AJCC cancer staging manual, 2017.
We also described the changes of AJCC/UICC lung cancer staging systems as the study periods spans in the section of Discussion. Changes were marked with yellowish color in the text.
The 8th edition of the American Joint Committee on Cancer (AJCC) and Union for International Cancer Control (UICC) lung cancer staging system has been in use since January 1, 2018. Consequently, we revised all the pathological and clinical stages of patients who underwent surgery before 2018 according to the new staging system. Although the 9th edition was launched in 2024, the T descriptions remain un-changed, and all of our study population were N0. Therefore, we did not revise the staging according to the new system.
Comment 4
: In the methods section, the authors describe post-surgery remaining tumor. This does not make sense, because IA lung cancer can be treated with surgery, without any tissue remaining. Please explain.
Answer 4
: We are sincerely grateful for the reviewer’s valuable comment. And we are terribly sorry for the mistakes. We were fully aware that the term postoperative tumor measurement did not make sense, and could be misunderstood that remnant tumor exists. We corrected the word post operatively into after surgery, and added phrases describing that we measure the tumor size from the excised lung cancer specimen according to the reviewer’s advice in 2.2 Pathologic examination in the section of the Materials and Methods. Changes were highlighted with yellowish color in the text.
Correction 4
: We corrected the misused words and appended phrases describing how we measure the tumor size from the excised lung specimens. Corrections were marked with yellowish color in the text.
Tumor sizes were measured in two ways. Before surgery, the largest diameter of total tumor size and the consolidative part were measured using chest computed tomographic imaging (chest CT). After surgery, the total tumor size and invasive tu-mor size (excluding the lepidic portion and including only the invasive component) were measured from surgically excised lung specimens by pathologists. The total tumor size was defined as the largest diameter of the tumor, including ground-glass opacity (GGO). The consolidation tumor ratio (CTR) was calculated as the ratio of the consolidative size to the total size, as measured by chest CT. The invasive tumor ratio represents the proportion of the invasive tumor size to the total tumor size.
Comment 5
: The authors have only attempted correlations with EGFR mutations. Please explain the rationale for not performing ALK or PD-L1 analysis as well.
Answer 5
: Thank you very much for the reviewer's invaluable comments. Genetic variations associated with the presence of STAS could be important for establishing precision treatment strategies. We were only able to examine the presence of EGFR mutations in our enrolled patients because the costs of comprehensive genetic examinations, such as next-generation sequencing for pathologically proven stage I lung cancer patients, are not reimbursed by the national insurance system. Based on the reviewer's advice, we believe it is important to describe the limited results of our genetic examinations.
We found several studies regarding genetic profiles in STAS-positive patients. In addition to wild-type EGFR mutations, KRAS mutations, ALK and ROS1 rearrangements, BRAF mutations, and wild-type HER2 have been reported to be related to the presence of STAS. However, we could not find any references linking the presence of STAS with PD-L1 expression. We have appended sentences describing the results of our study on genetic mutations along with the limitations of our research according to the reviewer’s advice. These changes are marked in yellow in the text.
Correction 5
: We added sentences describing the results our study on genetic mutations along with the limitations of our research in the section of Discussion as the reviewer had pointed out. Corrections were highlighted with yellowish color in the text.
The presence of STAS was associated with wild-type EGFR in our study, which is consistent with previous studies [3, 35, 36]. Although our data on molecular characteristics were limited to EGFR mutations, other molecular features such as KRAS mutation, ALK and ROS1 rearrangements, BRAF mutations, and wild-type HER2 have been reported to be related to the presence of STAS [1, 35, 37, 38]. Several studies have reported no significant relationship between STAS and EGFR mutations. [6, 10, 39].
:
Comment 6
: Please make sure to have a cross-check with clinical colleagues for consistency of terms.
Answer 6
: Thank you very much for your valuable feedback. In response to your suggestion, we have consulted with our clinical colleagues to ensure the consistency and accuracy of the terminology used in our manuscript. Their expertise has been crucial in verifying that all terms are in alignment with current clinical practice and standards. We have carefully revised the manuscript based on their input, ensuring that the terminology throughout is both consistent and appropriate. Should there be any specific terms or sections you would like us to review further, we would be happy to address them. We greatly appreciate your attention to this detail and your thoughtful suggestions.
The authors sincerely appreciate the editor’s and reviewers’ valuable comments. We carefully revised our manuscript according to the reviewers’ precious advice. We believe we did our best to improve the quality of our manuscript, and wish our revision have better achievement. If there were anything to be corrected or appended, please let us know and we promise we will make every effort to revise again.

Reviewer 4 Report
Comments and Suggestions for Authors
Surgical type including wedge, segmentectomy, or lobectomy is crucial for stage I NSCLC based on oncological principle.
In additional, visceral pleural invasion and neurovascular invasion can also make substantial impact on recurrence.
The aforementioned key factors should be incorporated into multivariable adjustment when considering recurrence analysis.
Comments on the Quality of English LanguageMinor editing of English language required
Author Response
RESPOND TO THE REVIEWERS
The authors would like to thank Reviewers for careful review of our manuscript and providing us with their comments and suggestion to improve the quality of the manuscript. The following responses have been prepared to address all of the editor’s comments in a point –by-point fashion.
Reviewer 4
Comment 1
: Surgical type including wedge, segmentectomy, or lobectomy is crucial for stage I NSCLC based on oncological principle.
Answer 1
: My coauthors and I are sincerely grateful for the reviewer's valuable comments. The extent of surgical resection based on oncological principles is crucial for stage I non-small cell lung cancers. All of our enrolled patients underwent Video-Assisted Thoracoscopic Surgery (VATS), and no open conversions were performed. Specifically, wedge resections were performed on 34 patients, segmentectomies on 19 patients, and lobectomies on 165 patients. There were no statistically significant differences between the STAS (-) and STAS (+) groups regarding surgical methods or tumor locations. Perioperative characteristics and comparisons between STAS (-), STAS (+)-L, and STAS (-)-L are described in the supplementary materials (Table S3). In accordance with the reviewer's advice, we have included additional sentences describing the surgical methods in the section of Discussion. These changes have been highlighted in yellow in the text.
Correction 1
: We appended sentences describing surgical methods in the section of Discussion as the reviewer’s advice. Changes were marked with yellowish color in the text.
All of the enrolled patient underwent VATS (Video-assisted thoracoscopic surgery) procedures, no robotic assisted procedures were performed, and no open conversion were enforced. The extents of surgical resection along locations were described in Supplementary materials (Table S3). No statistical significances were found between STAS subgroups, therefore we did not refer the results.
Comment 2
: In additional, visceral pleural invasion and neurovascular invasion can also make substantial impact on recurrence.
Answer 2
: We sincerely appreciate the reviewer’s insightful comments. The presence of visceral pleural invasion and neurovascular invasion has indeed been reported to significantly impact recurrence in early-stage lung adenocarcinoma. We recognize the importance of including these factors in recurrence risk analyses. Numerous studies investigating risk factors for STAS have frequently associated these factors with the presence of STAS. However, in our dataset, these factors did not show statistical significance in univariate analyses for both recurrence and the presence of STAS (Table S7 and Table 5). We believe this lack of significance may be attributed to the relatively small study population in our research. In response to the reviewer’s suggestion, we have added statements to the section of Discussion, highlighting these limitations. The corrections have been marked in the text with a yellow highlight.
Correction 2
: We appended sentences describing those points as limitations in the section of Discussion according to the reviewer’s comment. Changes were highlighted with yellowish color in the text.
The small study population might be affective to the recurrence risk factor analysis. Visceral pleural invasion along with the presence of lymphatic, vascular invasion has been frequently referred as recurrence risk factors for stage I lung adenocarcinomas [40-42]. In our study, no aforementioned factors seemed to have significance both in uni- and multivariate analysis (Supplementary materials, Table S7).
[40] Nitadori, J.-i., et al., Impact of micropapillary histologic subtype in selecting limited resection vs lobectomy for lung adenocarcinoma of 2cm or smaller. Journal of the National Cancer Institute, 2013. 105(16): p. 1212-1220.
[41] Nentwich, M.F., et al., Lymphatic invasion predicts survival in patients with early node-negative non–small cell lung cancer. The Journal of thoracic and cardiovascular surgery, 2013. 146(4): p. 781-787.
[42] Lakha, S., et al., Prognostic significance of visceral pleural involvement in early-stage lung cancer. Chest, 2014. 146(6): p. 1619-1626.
Comment 3
: The aforementioned key factors should be incorporated into multivariable adjustment when considering recurrence analysis.
Answer 3
: Thank you very much for the reviewer’ precious advice. The presence of visceral pleural invasion and neurovascular or lymphatic invasion are important risk factors for recurrence in lung adenocarcinomas, they should be incorporated into multivariable adjustment for recurrence analysis. In our data set, those factors showed no statistical significance for recurrence risk factor analysis. We added results uni- and multivariate recurrence risk analysis as a supplementary material (Table S7). Cox-proportional Hazard methods were used. Changes were marked with yellowish color in the text.
Correction 3
: We have added results of recurrence risk factor analysis as a a supplementary material (Table S7). Corrections were Highlighted with yellowish color in the text.
Table S7. Recurrence risk factor analysis. Cox-proportional hazard methods were used.
|
Variables |
Univariate |
Multivariate |
||||||
|
p-value |
Odds ratio |
95% CI |
p-value |
Odds ratio |
95% CI |
|||
|
Age |
0.730 |
1.2 |
(0.389–3.858) |
|||||
|
Sex, male |
0.166 |
0.4 |
(0.141–1.402) |
|||||
|
Consolidative size |
0.034 |
1.1 |
(1.010–1.296) |
0.608 |
1.1 |
(0.803–1.454) |
||
|
Invasive tumor size |
0.044 |
1.2 |
(1.004–1.339) |
0.764 |
1.0 |
(0.757–1.227) |
||
|
Micropapillary pattern |
0.000 |
9.9 |
(2.753–35.470) |
0.007 |
6.6 |
(1.688–26.116) |
||
|
STAS (+)-H |
0.001 |
7.9 |
(2.470–25.551) |
0.030 |
3.6 |
(1.132–11.470) |
||
|
Alveolar spaces >3 |
0.013 |
4.3 |
(1.370–13.657) |
0.734 |
0.7 |
(0.119–4.479) |
||
|
Smoking history |
||||||||
|
Ex >15 |
0.034 |
6.3 |
(1.145–34.179) |
0.113 |
4.2 |
(0.711–24.973) |
||
Abbreviations: CI, confidence interval; STAS, spread through air spaces
The authors sincerely appreciate the editor’s and reviewers’ valuable comments. We carefully revised our manuscript according to the reviewers’ precious advice. We believe we did our best to improve the quality of our manuscript, and wish our revision have better achievement. If there were anything to be corrected or appended, please let us know and we promise we will make every effort to revise again.

Round 2
Reviewer 2 Report
Comments and Suggestions for Authors
Thank you for your revision. Most of my questions were addressed accordingly. However, the most important one is how did you choose the right slide, which absolutely relating to the pathology accuracy, were not responded satisfactorily. It does not depend on how you cut the specimens. In the modern pathology, I think whole slide imaging is the most scientific procedure.
Author Response
RESPOND TO THE REVIEWERS (Round 2)
The authors would like to express our sincere gratitude for your thorough review of our manuscript and for providing insightful comments and suggestions to enhance its quality. We have carefully considered all of your feedback and have prepared the following responses to address each of your comments in a point-by-point manner. Thank you once again for your valuable advice.
Reviewer 2
Comment 1
: Thank you for your revision. Most of my questions were addressed accordingly. However, the most important one is how did you choose the right slide, which absolutely relating to the pathology accuracy, were not responded satisfactorily. It does not depend on how you cut the specimens. In the modern pathology, I think whole slide imaging is the most scientific procedure.
Answer 1
: I and my coauthors would like to express our sincere gratitude for your careful comment. And we would like to apologize for not sufficiently addressing your previous question; How much time you had to spend on measuring the STAS farthest distance and the number of alveolar spaces in each case?
As the reviewer had pointed out, whole slides should be carefully examined to determine the distance between tumor margin and the farthest STAS. We had reviewed all pathologic slides of all patients according to the reviewer’s comments. The slides had been made following standard processes after surgical resection, precise gross inspection, measurement, photography, sectioning and histologic processing by expert pathologists.
The lung specimens were cautiously sliced into thin section ensuring all areas of interest including tumor and surrounding tissues were adequately comprised. Detailed mappings were created indicating the location of tumor, margin and relevant microstructure. We would like to assure that we had examined all slides of Hematoxylin and Eosin (H&E) Staining, immunohistochemistry stains from each individual case for deciding the distance between tumor margin and STAS. We surely understand that whole slide imaging should be examined as the reviewer had mentioned. The methodology we had used in our study could have limitation and shortcomings. We would like to append this limitation of our study in the section of Discussion, as the reviewer’s comment. Corrections were marked with yellowish color in the text.
Correction 1
: We added sentences the pitfall of our study as limitation in the end of the section of Discussion according to the reviewer’s advice. Changes were highlighted with yellowish color in the text.
Although the slides we examined were prepared following standard pathological protocols, including gross inspection, measurement, sectioning, and histological processing, it is possible that STAS could exist in relevant parenchyma that was excluded from sectioning and mapping. Future standard guidelines for detecting STAS, including differential diagnoses from artifacts through immunohistochemistry and molecular analyses, may provide useful tools for identifying this phenomenon.
The authors sincerely appreciate the editor's and reviewers' valuable comments. We have carefully revised our manuscript in accordance with the reviewers' insightful advice. We believe these revisions have significantly enhanced the quality of our manuscript. Should there be any further corrections or additions required, please inform us, and we will make every effort to address them promptly.

Reviewer 3 Report
Comments and Suggestions for Authors
I have seen the amendments in your manuscript and I hope my comments have improved your already nice work.
Because your study population is from 2012 until 2022 I commented about using the 8th classification of lung cancer for your patients, something that you did. But, I think there is no need to refer to the 9th classification in this paper.
Author Response
RESPOND TO THE REVIEWERS (Round 2)
The authors would like to express our sincere gratitude for your thorough review of our manuscript and for providing insightful comments and suggestions to enhance its quality. We have carefully considered all of your feedback and have prepared the following responses to address each of your comments in a point-by-point manner. Thank you once again for your valuable advice.
Reviewer 3
Comment 1
: I have seen the amendments in your manuscript and I hope my comments have improved your already nice work. Because your study population is from 2012 until 2022, I commented about using the 8th classification of lung cancer for your patients, something that you did. But, I think there is no need to refer to the 9th classification in this paper.
Answer 1
: I and my coauthors would like to express our sincere gratitude for your careful comment. And we would like to apologize for referring the 9th Classification of lung cancer staging. We would like to eliminate sentences relating with the 9th classification as well as the reference according to the reviewer’s advice.
Correction 1
: We removed sentences as well as the reference relating with the 9th classification of lung cancer staging in the end of the section of Discussion according to the reviewer’s comment.
Although the 9th edition was launched in 2024 [43], the T descriptions remain un-changed [43], and all of our study population were N0. Therefore, we did not revise the staging according to the new system.
[43] Rami-Porta, R., et al., The International Association for the Study of Lung Cancer Lung Cancer Staging Project: Proposals for Revision of the TNM Stage Groups in the Forthcoming (Ninth) Edition of the TNM Classification for Lung Cancer. Journal of Thoracic Oncology, 2024.
The authors sincerely appreciate the editor's and reviewers' valuable comments. We have carefully revised our manuscript in accordance with the reviewers' insightful advice. We believe these revisions have significantly enhanced the quality of our manuscript. Should there be any further corrections or additions required, please inform us, and we will make every effort to address them promptly.

Round 3
Reviewer 2 Report
Comments and Suggestions for Authors
I suggest the E-I-C to send your manuscript to pathologists if they can accept your explanation. Let the experts decide.